# Awareness and Knowledge Regarding the Consumption of Dietary Fiber and Its Relation to Self-Reported Health Status in an Adult Arab Population: A Cross-Sectional Study

**DOI:** 10.3390/ijerph17124226

**Published:** 2020-06-13

**Authors:** Hanan Alfawaz, Nasiruddin Khan, Haya Alhuthayli, Kaiser Wani, Muneerah A. Aljumah, Malak Nawaz Khan Khattak, Saad A. Alghanim, Nasser M. Al-Daghri

**Affiliations:** 1Department of Food Science & Nutrition, College of Food Science & Agriculture, King Saud University, Riyadh 11495, Saudi Arabia; h.f.h4@hotmail.com; 2Biochemistry Department, Chair for Biomarkers of Chronic Diseases, King Saud University, Riyadh 11451, Saudi Arabia; wani.kaiser@gmail.com (K.W.); malaknawaz@yahoo.com (M.N.K.K.); aldaghri2011@gmail.com (N.M.A.-D.); 3Department of Food Science and Human Nutrition, College of Applied and Health Sciences, A’ Sharqiyah University, Ibra 400, Oman; nasiruddin2006@gmail.com; 4Almaarefa University, College of Medicine Medical Student, Riyadh 11597, Saudi Arabia; aljumahmd@gmail.com; 5Department of Health Administration, Health and Hospital Administration Program, College of Business Administration, King Saud University, Riyadh 11352, Saudi Arabia; sagksu@gmail.com

**Keywords:** dietary fiber, nutrition knowledge, fiber and health, Saudi Arabia

## Abstract

The objective of this study was to examine the awareness, knowledge, and habits regarding dietary fiber intake and to analyze its relationship with self-reported health status among Saudi adults. A survey-based study using face-to-face interview was designed, and 1363 apparently healthy adult Saudi males and females participated. Most participants were females (81.2%), aged 25 and above (87.2%), and were educated at least up to the secondary level of education (80.8%). The majority of the participants were aware of the role of fiber-rich foods in health conditions such as obesity (70.5%), cardiovascular diseases (68.9%), and regulation of blood sugar (68.9%), with females significantly having higher nutrition knowledge than males. A disconnect in translating this nutrition knowledge was observed particularly in food choices when eating out, where preferences for white bread (84.4%), fried potatoes (69.9%) and peeled fruits (60.6%) were significantly higher than preferences for cooked vegetables (29.6%) and brown bread (18.1%). The most common reason for this disconnect was due to perception that foods rich in dietary fibers were expensive (72.1%), have less health benefits (56.5%), were not readily available (51.6%), and participants’ disliking of the taste (52.8%). Participants in the highest quartile (Q4) for dietary fiber consumption reported a lower prevalence of constipation (odds ratio, 95% confidence interval of 0.40, 0.28–0.57, *p* < 0.01), high cholesterol (0.43, 0.27–0.68, *p* < 0.01) and obesity (0.67, 0.44–0.98, *p* = 0.03) than participants in the lowest quartile (Q1). Dietary fiber intake appears to be protective against constipation, high cholesterol and obesity in Saudi adults. However, a disparity observed between knowledge and attitude towards intake of dietary fibers could limit its health benefits. Further studies including adolescents should be conducted to impart knowledge on the emotional, cognitive and sensory factors related to food choices in order to minimize the gap between nutrition knowledge and the consumption of healthy high-fiber diets.

## 1. Introduction

Balanced dietary habits and a healthy lifestyle play important roles in every stage of human life, and may prove beneficial for the prevention and even treatment of some diseases [1]. Young people and adults must be aware of the basic pros and cons of the food they eat. Dietary fiber (DF) is one such element in a balanced diet and refers to the edible portion of the heterogeneous mixture of plant foods polysaccharides and lignin, which are resistant to absorption and digestion in the small intestine [2,3]. They are analogous to carbohydrates, provide bulk in the diet and aid in proper gastrointestinal (GI) function [4]. The major food sources of DF includes wholegrain cereals, legumes, fruits and vegetables and their contribution to dietary fiber has been reported as 50%, 30–40%, and 16%, respectively [5]. Fiber content and composition differs based on different food sources and the amount of intake [6].

Epidemiological studies have shown the preventive role of DF in several diseases such as cardiovascular, cancer, obesity and type 2 diabetes [7,8]. The intake of a diet rich in fiber can lead to decreased blood pressure [9]; an increased rate of bile excretion, thus reducing total and LDL cholesterol (LDL-C) [10]; and increased insulin sensitivity [11]. It can also stimulate optimal gastrointestinal function and prevent certain disorders [12]. However, excess DF intake can produce abdominal distension, intestinal gas, bloating and cramping [13,14]. It is recommended that a high-fiber intake must always be accompanied with high fluid intake as it allows the absorption of water by the intestines and, therefore, produces a softer and bulkier bowel movement. Moreover, higher than recommended intake of DF can also limit absorption of some nutrients and make them less available to the body. For instance, an inhibiting effect of fiber had been demonstrated on non-heme iron absorption [15].

The populations from many Western countries have been reported to consume fiber below the recommended levels [6]. This pattern of DF intake can have deleterious health effects. Recommendations of total fiber intake for an average population are different in some countries and range from 20–45 g/day [16]. However, according to the dietary reference intakes (DRIs) developed by the Institute of Medicine of The National Academics, the adequate intake (AI) of dietary fibers for adult males and females aged 20–50 years is 30–38 g/day and 21 to 25 g/day, respectively; for children it is 19–31 g/day, while for pregnant and lactating women it is 28–29 g/day [17]. For elderly people (>50 years) and for those suffering from intestinal complications, individualized dietary guidance by a dietitian is needed, however, 30 g/day and 21 g/day of DF, respectively, for healthy older men and women is generally recommended [18]. The plausible causes for this unhealthy fiber intake pattern may include the reduced intake of wholemeal products in daily diets and higher prices along with unavailability of a varied range of cereal products [19,20]. A study performed by Ahmed and colleagues in rural and urban areas of Bangladesh demonstrated the requirement of awareness and consumer education in order to increase DF intake [21]. A recent survey from Croatia demonstrated the importance of nutritional knowledge of healthy foods. The study reported that the knowledge of fiber consumption varied depending on age, gender, education level, and location (rural versus urban) [22]. Similar results from Romania demonstrated the vital role of knowledge, information and community intervention regarding fiber intake and its role in health status [23]. Thus, nutrition education and health awareness programs are preferred choices to educate societies to develop proper dietary habits.

Saudi Arabia has witnessed a rapid inclination from its traditional diet towards Western diets and lifestyles [24]. Studies from several parts of Saudi Arabia showed low levels of healthy diet awareness, sedentary behavior, imbalanced nutrient intake and unhealthy dietary patterns including increased use of high levels of carbohydrates, fat, free sugars, sodium, cholesterol and low level of dietary fiber [25,26]. This sudden change in dietary behavior has been reported as a major rationale for the emergence of several diseases in Saudi Arabia, including obesity, diabetes, dyslipidemia and coronary heart disease [27,28]. Since DF is an essential part of a balanced diet, lack of evidence on the knowledge, awareness and attitude towards the role of DF, especially in this part of the world, has led the authors to investigate on this subject. The present study aimed to the adult Saudi males and females’ awareness and knowledge of the role of DF and its relationship with their self-reported health status.

## 2. Materials and Methods

### 2.1. Study Design and Participants

This survey based cross-sectional study was designed and conducted at the capital Riyadh, as a representation of the urban population of Saudi Arabia. The inclusion criteria were (1) Saudi adult males and females aged between 20–70 years who responded to the announcement from January–May 2018, (2) the ability to comprehend properly relevant information. Exclusion criteria included non-Saudi individuals, children, adolescents, and pregnant women. The interviews were conducted by trained professionals who interviewed and asked the participants a series of questions from the validated questionnaire and noted their responses. Areas of recruitment included schools, colleges, shopping malls and public parks across Riyadh which were selected in a cluster random sampling method during the course of the survey. The total participants who responded included 1363 apparently healthy adults (N = 256 males, 18.8%; N = 1107 females, 81.2%). The response data from all these participants were compiled, analyzed and presented in this study.

All participants gave their informed consent before inclusion. The study was conducted in accordance with the Declaration of Helsinki, and the protocol including the questionnaires used in this survey was approved by the Ethics Committee for Scientific Research and Post Graduate Studies at the College of Science, King Saud University, Saudi Arabia (reference# 4/67/175981).

### 2.2. Questionnaires, Data Collection and Measurements

A pilot study including 50 participants was performed to confirm the reliability and validity of the questionnaire which was developed to investigated consumption, awareness, and knowledge of the importance of dietary fibers in the Saudi Adult population. Content and face validity were completed to clarify all the questions. The questionnaire was then reviewed by experts in the related fields. Moreover, external reviewers provided their feedback and opinion in developing/improving the questionnaire. Expert feedback and suggestions were incorporated in the final questionnaire. Reliability test was conducted using Cronbach’s α coefficient which yielded >70% for each section of the questionnaire used in this study. The final form of the questionnaire used in this survey study and the Cronbach’s alpha coefficient reliability test are presented separately (Appendix A, respectively).

The questionnaire used in this survey was divided into five sections:Socio-demographic characteristics including sex, age, marital status, family income, educational qualification, etc.Consumption of DF sources where participants had to indicate how often he or she consumes a selected list (30 items) of fiber-rich foods. The options were “don’t consume”, “less than once a week”, “1–2 times a week”, “≥3 times a week”, “daily”. Data were presented in the table as the frequency of participants consuming these food items regularly (≥3 times per week), rarely (twice or less a week) and those who do not consume at all.Knowledge about DF and their health impacts, and the preference of food choices while eating out/having fast foods. The interviewer assessed the knowledge of the participants by asking about the relation of DF in health conditions like cardiovascular diseases, obesity, diabetes, high cholesterol etc.; sources of DF in food; and the preference of food items while eating out.Reasons for the lack of DF in their diet. The participants had to answer whether they thought foods rich in DF were expensive; or foods rich in DF had limited health benefits; or whether they did not like the taste; or whether they had a perception of them not being readily available.Self-reported health status of the study participants, including diabetes, obesity, high cholesterol, hypertension etc.

### 2.3. Statistical Analysis

Data were analyzed using the SPSS 22.0 (SPSS Inc., Chicago, IL, USA). Data were presented as frequencies (%). Chi-square test was used to examine gender differences in socio-demographic characteristics; regular consumption of fiber-rich foods; knowledge, preference, and attitude towards DF. The participants were scaled as 0–30 and divided into four quartiles based on whether they reported regular consumption of the 30 listed fiber-rich food items or not. The self-reported health status of the participants was presented as frequencies (%) in the four DF intake quartiles. Multinomial regression analysis was conducted and the data were presented as odds ratio (O.R.) with a 95% confidence interval (95% CI) representing the odds of having the disease in different fiber intake quartiles compared to the lowest quartile. The O.R. was adjusted for age and other socio-economic parameters like social status, income and education. All *p*-values were two-tailed, and *p*-values < 0.05 were considered significant.

## 3. Results

### 3.1. Socio-Demographic Characteristics of the Study Participants

Table 1 represents the demographic characteristics of the participants and their frequencies. Most of the participants were in the age-group of 26–55 (75.4% for males and 82.8% for females) and married (78.9% for males and 74.7% for females). Most of the participants were educated at least up to the secondary level of education (98.4% for males and 96.1% for females).

### 3.2. Consumption of Fiber-Rich Foods by the Study Participants

Table 2 shows the consumption of fiber-rich foods (≥3 times a week) among the study participants. The participants preferred fruits and vegetables mostly in the form of salad, cooked vegetables and fruit juices, while oats and wheat were the preferred cereals. Peanuts, pistachios and almonds were the preferred forms of nuts; dried pineapple, dried dates and raisins were the preferred forms of dried fruits; while in legumes, fava beans and yellow lentils were consumed more than Libya beans. The food choices were also influenced by gender.

### 3.3. Knowledge about Dietary Fibers in Study Participants

Table 3 presents the knowledge of the study participants about DF and their health effects, as well as their preferences regarding foods high in DF when eating out. The study participants were well aware of role of DF in the prevention of obesity and cardiovascular diseases. However, they seem to be less aware of the role of DF intake in flatulence and constipation. An amount of 59.4% of the participants responded correctly to the daily requirement of DF. More females responded correctly to these questions than males.

An amount of 47.8% of the study participants preferred regularly eating out or consuming fast food, and this particular result was comparable between genders. When eating out or having fast foods, they preferred to have white bread, potato and peeled fruits rather than brown bread and cooked vegetables. This preference was also gender-influenced.

### 3.4. Perceptions about Foods Rich in DF

Table 4 shows the reasons for limited DF consumption in the study participants. The four most important reasons cited in order of highest to lowest prevalence were that they thought foods rich in DF were expensive; that foods rich in DF have no beneficial health values; that they do not like the taste of foods rich in DF; and they thought foods rich in DF were not easily available. Gender differences were also evident in the proportions of participants citing these reasons.

Figure 1a summarizes the knowledge in the study participants of the role of DF in health; Figure 1b shows preferences of food items while eating out; and Figure 1c illustrates perceptions related to foods rich in DF.

### 3.5. Self-Reported Health Status of the Study Participants and Its Association with DF Intake

Table 5 provides insight into the self-reported health status of the study participants and its association with DF intake. The most common were constipation (28.9%), high cholesterol (14.6%) and obesity (14.4); while the least common were cancer (1.2%) and cardiovascular disease (1.9%). A significantly higher percentage of males than females exhibited conditions such as diabetes (13.7% vs. 8.5%, *p* < 0.011), hypertension (15.2% vs. 8.5%, *p* < 0.001), and high cholesterol (21.5% vs. 13%, *p* < 0.001). On the other hand, more females than males had constipation (30.1% vs. 23.6%, *p* < 0.04), and osteoporosis (14.1% vs. 5.3%, *p* < 0.001).

Constipation, high cholesterol and obesity was found to be significantly and inversely correlated with the DF intake (based on the regular consumption of 30 listed fiber-rich food items). The individuals in the highest dietary intake quartile (Q4) showed 60%, 57% and 33% less risk, respectively, for constipation, high cholesterol and obesity when compared to those in the lowest quartile (Q1). These observations were more or less similar when the data were divided between genders. The odds ratios were adjusted for age and other socio-demographic variables such as social status, income and education.

Figure 2 summarizes the results of Table 5 for all participants (a), males (b) and females (c).

## 4. Discussion

The present survey-based study highlighted the level of knowledge, awareness and attitude towards the consumption of foods rich in DF and its health benefits among a Saudi adult population. The study also presented the association of the self-reported health conditions of the participants with the regular intake of food items rich in DF. This study reveals a discrepancy between the level of knowledge of the importance of DF in a balanced healthy diet and the food choices of the study participants, especially when eating out. The study also enlists the reasons for not adding DF in the regular diet in a Saudi population. This study also showed that health conditions—especially constipation, high cholesterol and obesity—were significantly and inversely associated with the consumption of DF.

Health benefits such as reduced risk of obesity, type 2 diabetes etc. could be derived by consuming the recommended amount of DF in our diet [7,8]. A fiber intake of at least 25 g/day was recommended by World Health Organization (WHO) [6], which may vary slightly based on different populations. However, the average daily intake of less than 10 g/day of DF in a regular Saudi diet was reported by our recent study [26], and this is very low when compared to the recommended levels which compromise the health benefits of a balanced high-fiber diet. Dietary fiber amount and composition differs in different food types, among which cereals comprises the major source of DF [29]. In our study, among cereals, wheat and maize are commonly consumed, while millet and quinoa were least consumed. Wheat is an important ingredient in the Saudi diet, and it is mostly consumed in the form of flat bread, local hamburgers (Samoli), pizza, etc. In our study, 47.8% (N = 652) of the study participants responded that they preferred eating out regularly, and 84.4% (N = 1150) preferred having white bread instead of brown bread (18.1%, N = 246) when eating out. Vegetables and fruits were the second important source of DF and participants preferred fried potatoes and peeled fruits instead of cooked vegetables and fruits with peels.

A recent study exhibited gender difference regarding health and immune function based on DF intake. The same study also showed a significantly higher consumption of cereals and grain products among males than females [30]. These findings corroborate our present study showing the same consumption pattern. Our present study demonstrated that females consume a higher frequency of consuming cooked vegetables than males. Riediger and colleagues also demonstrated that females are likely more consistent in consuming recommended amount of fruit and vegetables when compared to males [31]. Moreover, this trend was strongly supported by studies showing the increased participation of females in family food and maintaining their self-image and appearance [32,33]. Additionally, in the present study, the consumption of fruit juice, fresh peeled fruits, and cereals were higher for males when compared to females. Our result is supported by a study performed by Bagordo and colleagues that showed bread, fresh fruit and raw vegetables as the most frequently-consumed foods among males when compared to females [34].

Knowledge of nutrition and its effects is vital to maintain a healthy lifestyle, and it is a great tool for selecting healthy food [35]. In our study, the participants had fair knowledge about the benefits of foods rich in DF on the prevention of metabolic diseases like obesity (84.3%), cardiovascular diseases (70.5%) and regulation of blood sugars (68.9%). However, there was a discrepancy in its translation to healthy food choices, especially when eating out. N = 1150 (84.4%) of the participants preferred eating white bread instead of the whole bread (18.1%, N = 246) when eating out. This is in contrast with some studies where nutrition knowledge was found to be correlated with the actual food choices [36,37], and it may be in line with some other studies where food choices were seen as not only dependent on nutrition knowledge but also on external factors such as sensory evaluation, packaging, labelling, consumer perceptions, etc. [38,39]. In this study also, the most common reasons for not consuming foods rich in DF were the participants’ perception that they are expensive (72.1%), that they have less beneficial health effects (56.5%), and that they are not readily available (51.6%), as well their dislike for the taste of foods rich in DF (52.8%). Unfortunately, even though there is a greater dissemination of knowledge on healthy food choices through advertisements, print media, internet etc., there seems to be a great conflict of health and taste motives in the food choices [40]. Consumers often feel a dilemma in their food choices within healthy and tasty options, where they demand healthy food on a conscious cognitive level and based on their nutrition knowledge, but they give up and choose less healthy options based on a sensory level. A new methodical approach should be devised that combines both perspectives of imparting knowledge to overcome such consumer perceptions related to healthy=expensive=not easily available food choices, while at the same time working on ways to replace low fibrous ingredients with high fibrous ones without compromising much in regard to the sensory aspect.

In our research, significant differences were reported on the level of nutrition knowledge related to DF and health in both genders. Females seem to be more aware of the relation between dietary fiber consumption and prevention of metabolic diseases such as obesity, type 2 diabetes and overall health status (Table 3). This is in line with study of Tarcea et al. [23] which demonstrated more females than males responding correctly to the favorable role of fiber in cardiovascular safety, regulating blood sugar, reducing risk for obesity, etc. This supports our results, which may mainly be attributed to the greater involvement of females in food choice for their family [32] and a greater care for their own appearance [33]. Furthermore, females in the present study reported significantly higher rates than men of health conditions such as constipation. The reason behind this pattern encompasses many factors that may overcome the effect of knowledge. In USA, females are 2.2 times more likely to report constipation than males [41] due to hormonal factors, the effect of progesterone, and damage to the pelvic floor muscles. Such studies corroborate our present finding that in spite of sufficient knowledge and education about role of DF in health, females are more likely to have constipation than males.

The majority of male participants in our present study were more uncertain about the favorable role of DF in obesity, reduction in blood sugar, hypertension, and high cholesterol levels than their female counterparts. The lack of knowledge among male participants about these diabetes-specific risk factors may be associated with their significantly higher unhealthy statuses such as having diabetes, hypertension, and high cholesterol when compared to females. Recent studies in Arabian Gulf states have reported a rapid increase in type 2 diabetes mellitus and obesity in adults among the Saudi population. The prevalence of diabetes increased from 10.6% in 1989 to 32.1% in 2009 with a faster rate among Saudi men than women [42]. The increasing trends of this diseases have been mainly attributed to unhealthy diets and lifestyles. 

The major significant health problems among our participants were constipation, high cholesterol, obesity and osteoporosis. Moreover, in our study, females reported a significantly higher rate of osteoporosis than males, which is in accordance with an epidemiological analysis on 24 studies, where 34% and 30.7% of healthy Saudi women and men respectively aged 50–79 years had osteoporosis [43]. In our study, the self-reported health conditions—especially constipation, high cholesterol and obesity—were found to be significantly lower in those in the highest quartiles of DF intake (Table 5). A meta-analysis [44] showed that DF treatment improved stool frequency compared to placebo. Several studies, both observational [45] as well as randomized controlled trial [46] results, suggested that DF could be used to complement statins in reducing the circulating levels of total and LDL-cholesterol [47]. Similarly, studies showed that the consumption of DF could help in reducing the prevalence of obesity and metabolic syndrome by promoting the effect of good colon bacterium [48]. These studies and many others corroborate the results in our study for Saudi adult. Apart from these, DF consumption has also been linked with less prevalence of other diseases such as cardiovascular diseases, diabetes, gastric and pancreatic cancer, etc. [49]. In our study also, less prevalence could be seen in participants in the higher quartiles of DF intake (Table 5). However, the ORs are not statistically significant for the trend. The authors reason that this because of low overall prevalence of these health conditions in our study population and suggest further studies focusing on the impact of DF intake on the prevalence of these health conditions in Saudi adults.

The study has some limitations which the authors acknowledge here. This study was not designed to collect the dietary recall data and hence the actual average consumption of DF per day was not calculated. However, the reference of another of our published studies, where less than10g/day of DF was reported, was utilized to propose low intake of DF in a general Saudi diet. As children and adolescents were not included, a separate study on the dietary preferences and knowledge of DF in relation to health in Saudi children and adolescents may add more insight into the observations noted in this study. A major representation of our participants were educated at least up to the secondary level (96.6%) and were from families with at least average monthly income of 5000 Saudi Riyals (80.8%) and may not necessarily represent the illiterate Saudi population and those who come from low-income families. Furthermore, since 81.2% of our participants were female, the observations drawn on gender differences for knowledge and attitude towards DF should be cautiously inferred. 

## 5. Conclusions

The present study observes a discrepancy between knowledge and consumption of DF in Saudi adults. The study also suggests a significant inverse correlation between the consumption of DF and health conditions—especially constipation, high cholesterol and obesity. Nutritional education focusing on changing the perception towards foods rich in DF should be considered as a preferred choice to disseminate DF intake information in order to improve healthy lifestyle. Finally, more intervention studies including adolescents should be completed to impart knowledge on the emotional, cognitive and sensory factors related to food choices to minimize the gap between nutrition knowledge and the consumption of healthy high-fiber diets.

## Figures and Tables

**Figure 1 ijerph-17-04226-f001:**
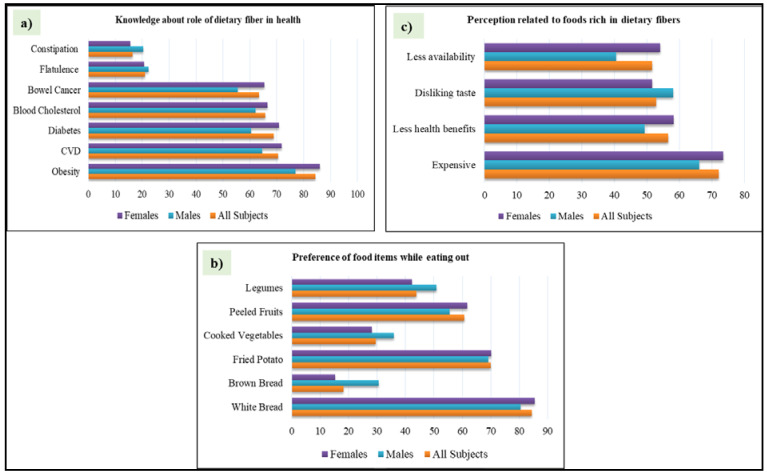
Knowledge (**a**), preferences (**b**) and perception (**c**) of study participants regarding foods rich in DF. Note: Data are presented in bar-graphs showing percentages of all participants, males alone and females alone.

**Figure 2 ijerph-17-04226-f002:**
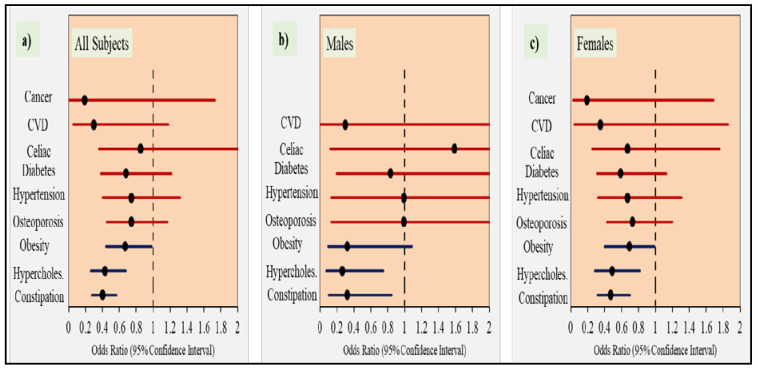
Odds of having the self-reported diseases in the highest quartile of DF consumption compared to lowest quartile. Note: Data are presented as O.R. (odds ratio, black dot) and 95% CI (confidence interval, horizontal line) represents the odds of having the self-reported diseases in those in the highest quartile of DF consumption (based on whether or not study participants regularly consume the listed 30 food items rich in DF) compared to those in the lowest quartile. The vertical dotted line represents the reference (lowest quartile). The data is presented for all participants (**a**), males (**b**), and females (**c**).

**Table 1 ijerph-17-04226-t001:** Socio-demographic characteristics of the study participants.

**Parameters**	**All**	**Male**	**Female**	***p***
N	1363	256	1107	
**Age Group (Years)**				**<0.01**
20–24	176 (12.9)	25 (9.8)	151 (13.6)	
25–35	411 (30.2)	70 (27.3)	341 (30.8)	
36–45	457 (33.5)	72 (28.1)	385 (34.8)	
46–55	242 (17.8)	51 (19.9)	191 (17.3)	
56–65	76 (5.6)	37 (14.5)	39 (3.5)	
66–70	1 (0.1)	1 (0.4)	0 (0.0)	
**Marital Status**				**0.02**
Married	1029 (75.5)	202 (78.9)	827 (74.7)	
Unmarried	262 (19.2)	50 (19.5)	212 (19.2)	
Widow	47 (3.4)	1 (0.4)	46 (4.2)	
Divorce	25 (1.8)	3 (1.2)	22 (2.0)	
**Family Income (SAR/month)**				**<0.01**
Low (<5000)	262 (19.2)	62 (24.2)	200 (18.1)	
Average (5000–10,000)	444 (32.6)	51 (19.9)	393 (35.5)	
Moderate (10,001–16,000)	384 (28.2)	55 (21.5)	329 (29.7)	
High (>16,000)	273 (20.0)	88 (34.4)	185 (16.7)	
**Education Level**				0.26
Read and Write	4 (0.3)	1 (0.4)	3 (0.3)	
Primary	1 (0.1)	0 (0.0)	1 (0.1)	
Intermediate	42 (3.1)	3 (1.2)	39 (3.5)	
Secondary	166 (12.2)	36 (14.1)	130 (11.7)	
Diploma	168 (12.3)	34 (13.3)	134 (12.1)	
Graduate	860 (63.1)	153 (59.8)	707 (63.9)	
Post-Graduate	122 (9.0)	29 (11.3)	93 (8.4)	

**Note:** Data represented as N (%). Differences in these characteristics between genders were calculated by Pearson chi-square test and depicted as *p*-value. *p*-value < 0.05 was considered statistically significant difference (bold).

**Table 2 ijerph-17-04226-t002:** Consumption of fiber-rich foods and supplements among participants.

	All			Males			Females			*p*
	Yes	No	Rarely	Yes	No	Rarely	Yes	No	Rarely	
**Fiber Supplements**	288 (21.1)	1074 (78.8)		61 (23.8)	195 (76.2)		227 (20.5)	879 (79.4)		**<0.001**
**Fruits and Vegetables**										
Salad	1184 (86.9)	23 (1.7)	156 (11.4)	222 (86.7)	8 (3.1)	26 (10.2)	962 (86.9)	15 (1.4)	130 (11.7)	0.115
Cooked Vegetables	1111 (81.5)	30 (2.2)	222 (16.3)	197 (77.0)	13 (5.1)	46 (18.0)	914 (82.6)	17 (1.5)	176 (15.9)	**0.001**
Fruit Juice	863 (63.3)	79 (5.8)	421 (30.9)	182 (71.1)	16 (6.3)	58 (22.7)	681 (61.5)	63 (5.7)	363 (32.8)	**0.007**
Fresh Peeled Vegetables	838 (61.5)	205 (15.0)	320 (23.5)	166 (64.8)	45 (17.6)	45 (17.6)	672 (60.7)	160 (14.5)	275 (24.8)	**0.037**
Fruits with peel	833 (61.1)	232 (17.0)	298 (21.9)	162 (63.3)	43 (16.8)	51 (19.9)	671 (60.6)	189 (17.1)	247 (22.3)	0.672
**Cereals**										
Oats	905 (66.4)	121 (8.9)	337 (24.7)	150 (58.6)	37 (14.5)	69 (27.0)	755 (68.2)	84 (7.6)	268 (24.2)	**0.001**
Whole Wheat	884 (64.9)	157 (11.5)	322 (23.6)	168 (65.6)	33 (12.9)	55 (21.5)	716 (64.7)	124 (11.2)	267 (24.1)	0.536
Refined wheat	861 (63.2)	157 (11.5)	345 (25.3)	165 (64.5)	32 (12.5)	59 (23.0)	696 (62.9)	125 (11.3)	286 (25.8)	0.684
Maize	572 (42.0)	246 (18.0)	545 (40.0)	97 (37.9)	57 (22.3)	102 (39.8)	475 (42.9)	189 (17.1)	443 (40.0)	0.113
Burghul	377 (27.7)	368 (27.0)	618 (45.3)	59 (23.0)	83 (32.4)	114 (44.5)	318 (28.7)	285 (25.7)	504 (45.5)	0.052
Barley	275 (20.2)	572 (42.0)	516 (37.9)	74 (28.9)	94 (36.7)	88 (34.4)	201 (18.2)	478 (43.2)	428 (38.7)	**0.001**
Millet	241 (17.7)	572 (42.0)	550 (40.4)	62 (24.2)	94 (36.7)	100 (39.1)	179 (16.2)	478 (43.2)	450 (40.7)	**0.007**
Quinoa	213 (15.6)	698 (51.2)	452 (33.2)	39 (15.2)	144 (56.3)	73 (28.5)	174 (15.7)	554 (50.0)	379 (34.2)	0.151
**Nuts**										
Peanuts	1015 (74.5)	61 (4.5)	287 (21.1)	198 (77.3)	11 (4.3)	47 (18.4)	817 (73.8)	50 (4.5)	240 (21.7)	0.525
Pistachios	982 (72.0)	89 (6.5)	292 (21.4)	191 (74.6)	19 (7.4)	46 (18.0)	791 (71.5)	70 (6.3)	246 (22.2)	0.257
Almonds	854 (62.7)	147 (10.8)	362 (26.6)	160 (62.5)	31 (12.1)	65 (25.4)	694 (62.7)	116 (10.5)	297 (26.8)	0.729
Walnuts	678 (49.7)	246 (18.0)	439 (32.2)	123 (48.0)	42 (16.4)	91 (35.5)	555 (50.1)	204 (18.4)	348 (31.4)	0.503
Hazelnut	581 (42.6)	288 (21.1)	494 (36.2)	125 (48.8)	49 (19.1)	82 (32.0)	456 (41.2)	239 (21.6)	412 (37.2)	0.069
Shea Seeds	241 (17.7)	738 (54.1)	384 (28.2)	49 (19.1)	145 (56.6)	62 (24.2)	192 (17.3)	593 (53.6)	322 (29.1)	0.193
**Dried Fruits**										
Dried Pineapple	627 (46.0)	237 (17.4)	499 (36.6)	125 (48.8)	38 (14.8)	93 (36.3)	502 (45.3)	199 (18.0)	406 (36.7)	0.459
Dried Dates	595 (43.7)	450 (33.0)	318 (23.3)	124 (48.4)	71 (27.7)	61 (23.8)	471 (42.5)	379 (34.2)	257 (23.2)	0.096
Raisin	573 (42.0)	313 (23.0)	477 (35.0)	129 (50.4)	42 (16.4)	85 (33.2)	444 (40.1)	271 (24.5)	392 (35.4)	**0.005**
Dried Figs	539 (39.5)	381 (28.0)	443 (32.5)	122 (47.7)	52 (20.3)	82 (32.0)	417 (37.7)	329 (29.7)	361 (32.6)	**0.001**
Apricot	433 (31.7)	434 (31.9)	496 (36.4)	97 (37.9)	66 (25.8)	93 (36.3)	336 (30.4)	368 (33.2)	403 (36.4)	**0.031**
**Legumes**										
Fava Beans	936 (68.7)	122 (9.0)	305 (22.4)	187 (73.0)	24 (9.4)	45 (17.6)	749 (67.7)	98 (8.9)	260 (23.5)	0.110
Yellow Lentils	868 (63.7)	147 (10.8)	348 (25.5)	157 (61.3)	32 (12.5)	67 (26.2)	711 (64.2)	115 (10.4)	281 (25.4)	0.495
Chickpeas	867 (63.6)	130 (9.5)	366 (26.9)	185 (72.3)	19 (7.4)	52 (20.3)	682 (61.6)	111 (10.0)	314 (28.4)	**0.009**
Green Beans	690 (50.6)	261 (19.1)	412 (30.2)	131 (51.2)	37 (14.5)	88 (34.4)	559 (50.5)	224 (20.2)	324 (29.3)	0.074
White Beans	380 (27.9)	464 (34.0)	519 (38.1)	90 (35.2)	59 (23.0)	107 (41.8)	290 (26.2)	405 (36.6)	412 (37.2)	**<0.001**
Libya beans	354 (26.0)	496 (36.4)	513 (37.6)	80 (31.3)	73 (28.5)	103 (40.2)	274 (24.8)	423 (38.2)	410 (37.0)	**0.004**

**Note:** Data represented as N (%). The data were divided into those who regularly consumed (≥3 times a week); rarely (twice or less a week) and those who said they do not consume at all. *p*-value < 0.05 was considered significant for differences between genders (bold).

**Table 3 ijerph-17-04226-t003:** Knowledge about dietary fibers in study participants.

	All (*N* = 1363)		Males (*N* = 256)		Females (*N* = 1107)		*p*
	Yes	No	Yes	No	Yes	No	
**Knowledge about DF**							
DF and prevention of obesity	1148 (84.3)	214 (15.7)	197 (77.0)	59 (23.0)	951 (86.0)	155 (14.0)	**<0.001**
DF and cardiovascular diseases	960 (70.5)	402 (29.5)	165 (64.5)	91 (35.5)	795 (71.9)	311 (28.1)	**0.004**
DF and regulation of blood sugar	939 (68.9)	423 (31.1)	155 (60.5)	101 (39.5)	784 (70.9)	322 (29.1)	**0.002**
DF and reduction in blood cholesterol	895 (65.7)	467 (34.3)	159 (62.1)	97 (37.9)	736 (66.5)	370 (33.4)	0.293
DF and prevention of bowel cancer	865 (63.5)	497 (36.5)	142 (55.5)	114 (44.5)	723 (65.4)	383 (34.6)	**<0.001**
DF and flatulence	286 (21.0)	1076 (78.9)	57 (22.3)	199 (77.7)	229 (20.7)	877 (79.2)	0.516
DF and constipation	223 (16.4)	1139 (83.6)	52 (20.3)	204 (79.9)	171 (15.5)	935 (84.5)	**0.020**
Too much DF and health	183 (13.4)	1179 (86.5)	42 (16.4)	214 (83.6)	141 (12.7)	965 (87.2)	0.058
DF daily intake requirement							0.060
5–10 g/day	151 (11.0)		27 (10.7)		124 (11.2)		
11–24 g/day	402 (29.6)		59 (23.0)		343 (31.0)		
25–38 g/day	810 (59.4)		170 (66.3)		640 (57.8)		
**Preference of DF intake in fast foods/eating out**							
Regularly eating fast foods/eating out	652 (47.8)	711 (52.2)	115 (44.9)	141(55.1)	536 (48.5)	570 (51.5)	0.552
Use of white bread in fast foods/eating out	1150 (84.4)	212 (15.6)	206 (80.5)	50 (19.5)	944 (85.4)	162 (14.6)	**0.011**
Use of fried potato in fast foods/eating out	952 (69.9)	410 (30.1)	177 (69.1)	79 (30.9)	775 (70.1)	331 (29.9)	0.09
Use of peeled fruits in fast foods/eating out	825 (60.6)	538 (39.4)	142 (55.5)	114 (44.5)	683 (61.7)	423 (38.2)	**0.001**
Use of legumes in fast foods/eating out	598 (43.8)	765 (56.1)	130 (50.8)	126 (49.2)	467 (42.2)	639 (57.8)	**0.012**
Use of cooked vegetables in fast foods/eating out	404 (29.6)	959 (70.4)	92 (35.9)	164 (64.1)	311 (28.1)	795 (71.9)	**0.005**
Use of brown bread in fast foods/eating out	246 (18.1)	1117 (81.9)	78 (30.5)	178 (69.5)	168 (15.2)	938 (84.8)	**<0.001**

**Note:** Data represented as N (%). DF is dietary fiber. *p*-value < 0.05 was considered significant for differences between genders (bold).

**Table 4 ijerph-17-04226-t004:** Reasons for the lack of DF in the diet.

	All (*N* = 1363)			Male (*N* = 256)			Female (*N* = 1107)			*p*
	Yes	No	Don’t Know	Yes	No	Don’t Know	Yes	No	Don’t Know	
•Foods rich in DF are expensive	983 (72.1)	215 (15.8)	165 (12.1)	169 (66.1)	48 (18.6)	39 (15.3)	814 (73.5)	167 (15.1)	126 (11.4)	0.07
•There is limited beneficial effect in consuming fiber-rich food to my health	770 (56.5)	478 (35.1)	115 (8.4)	126 (49.2)	95 (37.2)	35 (13.6)	644 (58.2)	383 (34.6)	80 (7.2)	**0.02**
•I do not like fiber-rich food taste	720 (52.8)	507 (37.2)	136 (10.0)	149 (58.1)	77 (30.1)	30 (11.9)	571 (51.6)	430 (38.8)	106 (9.6)	**0.04**
•Foods rich in DF are not easily available	703 (51.6)	478 (35.1)	180 (13.2)	104 (40.5)	104 (40.5)	48 (19.0)	601 (54.1)	374 (33.9)	132 (11.9)	**<0.01**

**Note:** Data represented as N (%). DF is dietary fiber. *p*-value < 0.05 was considered significant for differences between genders (bold).

**Table 5 ijerph-17-04226-t005:** Health status of the study participants according to the DF intake quartiles.

**All Participants (*N* = 1363)**									
**Fiber Intake Quartiles**	**Q 1 (340)**	**Q 2 (341)**	**Q 3 (341)**	**Q 4 (341)**	**Q1**	**Q2**	**Q3**	**Q4**	P^a^
Dietary fiber score	6 ± 2	12 ± 2	17 ± 1	23 ± 3	**Adjusted O.R. (95% C.I.)**				
	**% (N)**								
Constipation	37.2 (126)	29.6 (100)	29.2 (99)	19.1 (65)	Ref.	0.71 (0.51–0.98) *	0.70 (0.51–0.96) *	0.40 (0.28–0.57) **	**<0.001**
High cholesterol	20.4 (69)	11.8 (40)	15.3 (52)	10.9 (37)	Ref.	0.46 (0.29–0.73) **	0.66 (0.43–0.99) *	0.43 (0.27–0.68) **	**0.001**
Obesity	19.6 (66)	12.1 (41)	12.1 (41)	13.8 (47)	Ref.	0.57 (0.37–0.87) *	0.57 (0.37–0.87) *	0.67 (0.44–0.98) *	**0.027**
Osteoporosis	15.1 (51)	10.7 (36)	13.0 (44)	10.9 (37)	Ref.	0.68 (0.42–1.09)	0.94 (0.59–1.48)	0.74 (0.46–1.17)	0.307
Hypertension	9.5 (32)	12.1 (41)	10.3 (35)	7.3 (25)	Ref.	1.28 (0.76–2.16)	1.09 (0.64–1.87)	0.74 (0.42–1.32)	0.251
Diabetes	9.8 (33)	12.4 (42)	8.8 (30)	7.0 (24)	Ref.	1.21 (0.72–2.03)	0.87 (0.50–1.52)	0.68 (0.38–1.22)	0.225
Celiac disease	4.2 (14)	2.7 (9)	1.5 (5)	2.9 (10)	Ref.	0.78 (0.32–1.88)	0.46 (0.16–1.33)	0.85 (0.36–2.00)	0.507
CVD	2.7 (9)	1.2 (4)	2.9 (10)	0.9 (3)	Ref.	0.44 (0.13–1.54)	1.10 (0.42–2.95)	0.30 (0.08–1.18)	0.103
Cancer	1.8 (6)	0.9 (3)	1.8 (6)	0.3 (1)	Ref.	0.57 (0.13–2.45)	1.27 (0.37–4.37)	0.19 (0.02–1.73)	0.185
**Males (*N* = 256)**									
**Fiber intake quartiles**	**Q 1 (64)**	**Q 2 (64)**	**Q 3 (64)**	**Q 4 (64)**	**Q1**	**Q2**	**Q3**	**Q4**	P^a^
Dietary fiber score	6 ± 3	13 ± 2	17 ± 2	25 ± 3	**Adjusted O.R. (95% C.I.)**				
	**% (N)**								
Constipation	34.4 (22)	29.7 (19)	17.2 (11)	12.5 (8)	Ref.	0.83 (0.37–1.85)	0.39 (0.16–0.95) *	0.32 (0.12–0.85) *	**0.037**
High Cholesterol	28.2 (18)	25.0 (16)	20.3 (13)	12.5 (8)	Ref.	0.47 (0.19–1.18)	0.49 (0.19–1.27)	0.26 (0.09–0.75) *	0.076
Obesity	18.8 (12)	15.6 (10)	9.4 (6)	7.8 (5)	Ref.	0.79 (0.29–2.16)	0.42 (0.14–1.33)	0.32 (0.09–1.09)	0.198
Osteoporosis	7.8 (5)	6.3 (4)	0.0 (0)	6.3 (4)	Ref.	0.92 (0.21–3.98)	-	0.99 (0.22–4.49)	0.07
Hypertension	9.4 (6)	25.0 (16)	15.6 (10)	10.9 (7)	Ref.	3.74 (1.20–11.6) *	1.93 (0.58–6.40)	1.53 (0.43–5.39)	0.099
Diabetes	7.8 (5)	25.0 (16)	15.6 (10)	6.3 (4)	Ref.	3.37 (1.04–10.9) *	1.96 (0.56–6.79)	0.83 (0.19–3.56)	0.057
Celiac disease	1.6 (1)	4.7 (3)	1.6 (1)	3.1 (2)	Ref.	3.35 (0.31–36.2)	1.00 (0.06–18.4)	1.59 (0.12–21.7)	0.658
CVD	3.1 (2)	3.1 (2)	3.1 (2)	1.6 (1)	Ref.	1.12 (0.11–11.1)	1.16 (0.12–11.5)	0.30 (0.02–4.77)	0.697
Cancer	0.0 (0)	1.6 (1)	0.0 (0)	1.6 (1)	Ref.				
**Females (*N* = 1107)**									
**Fiber intake quartiles**	**Q 1 (277)**	**Q 2 (277)**	**Q 3 (277)**	**Q 4 (276)**	**Q1**	**Q2**	**Q3**	**Q4**	P^a^
Dietary fiber score	6 ± 2	12 ± 2	17 ± 1	23 ± 3					
	**% (N)**				**Adjusted O.R. (95% C.I.)**				
Constipation	37.3 (103)	30.1 (83)	30.9 (85)	21.6 (59)	Ref.	0.73 (0.51–0.99) *	0.81 (0.56–1.16)	0.47 (0.32–0.70) **	**0.001**
High cholesterol	18.2 (50)	9.8 (27)	13.8 (38)	10.2 (28)	Ref.	0.46 (0.27–0.78) **	0.70 (0.43–1.15)	0.49 (0.29–0.82) **	**0.010**
Obesity	20.1 (55)	11.2 (31)	12.7 (35)	15.0 (41)	Ref.	0.49 (0.31–0.81) **	0.59 (0.37–0.95) *	0.69 (0.40–0.99) *	**0.028**
Osteoporosis	17.2 (47)	11.2 (31)	15.6 (43)	12.4 (34)	Ref.	0.63 (0.38–1.05)	1.02 (0.63–1.64)	0.73 (0.44–1.20)	0.170
Hypertension	9.4 (26)	9.1 (25)	9.4 (26)	6.2 (21)	Ref.	0.99 (0.54–1.84)	1.09 (0.59–2.02)	0.67 (0.34–1.31)	0.472
Diabetes	10.5 (29)	9.4 (26)	7.6 (21)	6.6 (18)	Ref.	0.83 (0.46–1.52)	0.69 (0.37–1.31)	0.59 (0.31–1.13)	0.409
Celiac disease	4.7 (13)	2.2 (6)	1.8 (5)	2.6 (7)	Ref.	0.56 (0.20–1.56)	0.50 (0.17–1.48)	0.67 (0.25–1.76)	0.550
CVD	2.2 (6)	1.4 (4)	2.5 (7)	0.7 (2)	Ref.	0.70 (0.18–2.74)	1.18 (0.36–3.92)	0.35 (0.07–1.86)	0.396
Cancer	2.2 (6)	0.7 (2)	2.2 (5)	1.5 (1)	Ref.	0.36 (0.07–1.94)	1.15 (0.32–4.08)	0.19 (0.04–1.69)	0.191

**Note:** Data, represented by % (N) with self-reported diseases, are divided according to the dietary fiber intake quartiles calculated on the basis of whether the listed 30 food items were regularly consumed by the study participants or not. CVD is cardiovascular diseases. Odds ratio (O.R.) was calculated by multinomial regression and was reported as the odds of having the disease in the respective quartile compared to the first quartile. Data were adjusted for age and socio-economic status including income and education status and adjusted *p*-value (p^a^ as the trend was reported). P^a^ < 0.05 was considered significant (bold). Additionally, the O.R.s for individual quartiles against the reference (lowest quartile) were depicted as significant by * (*p* < 0.05) or ** (*p* < 0.01).

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
