# Peer review of "Awareness and Knowledge Regarding the Consumption of Dietary Fiber and Its Relation to Self-Reported Health Status in an Adult Arab Population: A Cross-Sectional Study"

_ijerph, 2020, doi:10.3390/ijerph17124226_

Round 1
Reviewer 1 Report
The menuscript regards the important issue of fiber-containing food preferences among the population living in Saudi Arabia. The results, though referring to the local community, relate to a society undergoing noticeable cultural and habitual changes. For these reasons, the results presented in the manuscript are interesting, important and should be published.
Several shortcomings need to be corrected before being published, listed below:
- Page 2, line 65-82: The authors describe recommended fiber intake, refer to other guidelines for children and adolescents and pregnant women. They do not write, however, whether and what are the possible recommendations for the elderly and those suffering from intestine diseases.
- Throughout the manuscript, the authors write about the population of Saudi Arabia, as a group that was surveyed, while the survey respondents were only residents of Riyadh. This should be clearly referred to in the description and discussion of the results.
- Page 5, line 154 and Table 2: Nuts are included in the dried fruit. Please describe them separately.
- Table 2: All product names must be in upper case.
- Table 3: In column II (All Yes) in the row concerning DF daily inatke requirement, the values should be moved one row below
- Table 3: Intake of fiber supplements - this row fits more with Table 2.
- Page 9, line 3 on this page (by the way, there is no line numbering here) rather “higher in knowing” than “higher in not knowing”, because Male/Yes it is 40.5%, Female/Yes it is 54.1%. The sentence needs to be rewritten to make it more understandable and logical.
- Table 4: The table title must be on a separate line.
- Table 4: First line: Food ... ia, or Foods ... are, not Food .... are
- Page 11, line 7: (by the way, the numbering of the lines must be continuous throughout the article), the authors described that the listed diseases are correlated with fiber consumption, while in fact they did not study the fiber consumption in grams and did not study the calorie equivalent of the entire diet, they only asked about eating specific groups of high fiber products. Such inference is therefore not justified.
- Line 14 – there is no line 14
- Figure 2: in part b) the first line for “cancer” is missing
- Page 15, line 30: Author's name is not “Natalie”, this is the first name.
Author Response
Response to Reviewer 1 Comments
The manuscript regards the important issue of fiber-containing food preferences among the population living in Saudi Arabia. The results, though referring to the local community, relate to a society undergoing noticeable cultural and habitual changes. For these reasons, the results presented in the manuscript are interesting, important and should be published.
Author Response: The authors would like to thank the reviewer for the valuable comments and suggestions that has greatly helped in improving the manuscript. The authors agree and appreciate the reviewer for recognizing the need to publish such data especially for communities who undergo noticeable cultural and habitual changes recently as pointed out.
Several shortcomings need to be corrected before being published, listed below:
1) Page 2, line 65-82: The authors describe recommended fiber intake, refer to other guidelines for children and adolescents and pregnant women. They do not write, however, whether and what are the possible recommendations for the elderly and those suffering from intestine diseases.
Author Response: The authors thank the reviewer for this suggestion. The authors would like to state that it is difficult to recommend the adequate dietary fiber intake for elderly people and especially those suffering from intestinal complications. An individualized dietary guidance by a dietitian is recommended for this group and this has been mentioned in the revised draft of the manuscript. Generally recommended values of 30 g/day and 21 g/day of dietary fiber respectively for healthy older men and women are also mentioned though in the revised draft.
2) Throughout the manuscript, the authors write about the population of Saudi Arabia, as a group that was surveyed, while the survey respondents were only residents of Riyadh. This should be clearly referred to in the description and discussion of the results.
Author Response: The authors agree with the reviewer that all the respondents from our study were from Riyadh city, the capital of Saudi Arabia. This was taken as a representation of the homogenous population of Saudi Arabia and this has been referred now in the revised manuscript (in the first line of section 2.1).
3) Page 5, line 154 and Table 2: Nuts are included in the dried fruit. Please describe them separately.
Author Response: The authors thank the reviewer for this suggestion. ‘Nuts’ have been separated from ‘dried fruits’ in the table as well as in the description.
4) Table 2: All product names must be in upper case.
Author Response: Thanks. All the product names in table 2 now start with the upper case in the revised manuscript.
5) Table 3: In column II (All Yes) in the row concerning DF daily intake requirement, the values should be moved one row below
Author Response: Thanks. This has been corrected in the revised manuscript.
6) Table 3: Intake of fiber supplements - this row fits more with Table 2.
Author Response: The authors agree with the reviewer and hence this piece of information has been adjusted in table 2 of revised manuscript.
7) Page 9, line 3 on this page (by the way, there is no line numbering here) rather “higher in knowing” than “higher in not knowing”, because Male/Yes it is 40.5%, Female/Yes it is 54.1%. The sentence needs to be rewritten to make it more understandable and logical.
Author Response: The description part in the entire result section has been revised to accommodate a comment from one of the reviewers and the information already present in the tables was omitted from the description section. In line with this, the whole sentence was rewritten in the revised draft.
Line numbering was added (continuous) in the revised draft.
8) Table 4: The table title must be on a separate line.
Author Response: Done in the revised draft.
9) Table 4: First line: Food ... ia, or Foods ... are, not Food .... are
Author Response: Thanks, corrected in the revised draft.
10) Page 11, line 7: (by the way, the numbering of the lines must be continuous throughout the article), the authors described that the listed diseases are correlated with fiber consumption, while in fact they did not study the fiber consumption in grams and did not study the calorie equivalent of the entire diet, they only asked about eating specific groups of high fiber products. Such inference is therefore not justified.
Author Response: The authors agree with the reviewer that the actual fiber consumption in grams by way of converting the diet to nutrient equivalents was not done as this was a qualitative survey based study. The subjects were in fact scaled as 0-30 and divided into four quartiles based on whether they reported regular consumption of the 30 listed fiber rich food items or not. The concerned statement has thus been revisited in the revised draft.
11) Line 14 – there is no line 14
Author Response: Thanks, this has been corrected by assigning continuous line numbers in the entire draft.
12) Figure 2: in part b) the first line for “cancer” is missing
Author Response: The authors thank the reviewer for this observation. The authors would like to state that since there was no male cancer patient in the lowest (reference, Q1) quartile (table 5), the odds of having this disease in the highest quartile (Q4) (plotted in figure 2) could not be generated and this has been indicated as dash sign (-) for the O.R.’s against this disease under males in table 5.
13) Page 15, line 30: Author's name is not “Natalie”, this is the first name.
Author Response: Thanks, this has been corrected.
Reviewer 2 Report
The manuscript is well written and the results provide a great deal of knowledge about the nutritional problems of the human population worldwide and particularly Saudi people. I agree with the authors, this work have to extend their study to other sectors of the population mainly children and adolescents. In addition, I recommend that the authors consider conducting a general medical examination of the study population to find out and corroborate if the participants really know their state of health, because the materials and methods section it is mentioned that an apparently healthy population was studied and later in the results different types of diseases are mentioned. Finally, although the results showed clear differences between male and female patients. The population has an important bias in that the number of males was three times less.
Author Response
Response to Reviewer 2 Comments
1) The manuscript is well written and the results provide a great deal of knowledge about the nutritional problems of the human population worldwide and particularly Saudi people. I agree with the authors, this work have to extend their study to other sectors of the population mainly children and adolescents. In addition, I recommend that the authors consider conducting a general medical examination of the study population to find out and corroborate if the participants really know their state of health, because the materials and methods section it is mentioned that an apparently healthy population was studied and later in the results different types of diseases are mentioned.
Author Response: The authors would like to thank the reviewer for the valuable comments and suggestions that has greatly helped in improving the manuscript. The authors agree and appreciate the reviewer for recognizing the need to publish such data especially for communities who undergo noticeable cultural and habitual changes and also to extend such studies to children and adolescents as nurturing healthy eating habits should start early in life. The authors welcome the recommendations from the reviewer to crosscheck the general medical examination of the study population; however, since the present study was a qualitative survey based research where the respondents were interviewed in colleges, shopping malls and public parks etc, its difficult, and cumbersome to trace back the respondents for further details. However, this suggestion could be used in future projects to design studies that could answer this research question.
2) Finally, although the results showed clear differences between male and female patients. The population has an important bias in that the number of males was three times less.
Author Response: The authors agree with the reviewer that female respondents were clearly more compared to male respondents. The primary objective, though, of the study was to investigate the awareness, knowledge and attitude of adult Saudis towards dietary fibers and its relationship with self-reported health status which was achieved by presenting the data separately for all subjects; males alone; and females alone. However, the authors agree that the inferences drawn due to differences in knowledge and attitude towards dietary fibers between the genders may have bias due to disproportionate numbers and this has been added as a limitation in the revised draft.
Reviewer 3 Report
Awareness, Knowledge and Consumption of Dietary Fiber and its Relation with Self-reported Health Status in Adult Arab Population
The authors have submitted a manuscript to analyse the awareness and knowledge on the role of dietary fibers and its relationship with the self-reported health status in adult Saudi males and females. However, the manuscript has some limitations that should be solved before the manuscript is to be considered for publication. My primary concern is the replicability of the research, the information reported by the researchers is not enough.
Title: please include the type of study carried out
Introduction:
page 2. Lines 44 - 48. References should be included to support the statements
The authors justify the lack of evidence for the study in adults:
“However, to the best of our knowledge, there is no study on adults in Saudi Arabia that reports the basic knowledge, awareness and attitude towards the role of dietary fibers in a balanced diet and its impact on the health status”.
Nevertheless, the authors have included young people (20 years) in the sample. Could you explain why?
2. Materials and Methods
the information included in this section is very brief and does not allow the study to be reproduced. Authors should provide more information on the following issues:
- The authors should report more information on the sample. Type of sampling, sample size and inclusion/exclusion criteria. In addition, the results section should show the number of participants recruited and how many of them were excluded from the study.
- The authors should show more information about the validation and reliability process (items). Also, authors should include the questionnaire as supplementary material. In addition, in the results section, should show the reliability and validity data of the questionnaire and the modifications proposed by the experts.
I think that the statistical analysis is adequate and allows to answer the research question.
3. Results
The results section is too long. I encourage authors not to repeat the information reported in the table.
I think it would be interesting to include in the table the statistically significant differences between genders of the characteristics shown in Table 1.
In line with previous comments (methods). I don't understand the concept rarely (Table 2). When is it considered rarely, once a week, once a month... Please, this answer to the questionnaire must be explained in the methodology.
Discussion
Line 4. Please, change correlation by association
Line 41. Please change ; to .
Line 62. Authors should unify sexes or gender throughout the text.
Although the authors have acknowledged the socio-demographic differences as a limitation of the study, I think that a logistic regression should have been conducted to control for these differences. Please let me know your opinion.
Author Response
Response to Reviewer 3 Comments
The authors have submitted a manuscript to analyze the awareness and knowledge on the role of dietary fibers and its relationship with the self-reported health status in adult Saudi males and females. However, the manuscript has some limitations that should be solved before the manuscript is to be considered for publication. My primary concern is the replicability of the research, the information reported by the researchers is not enough.
Author Response: The authors would like to thank the reviewer for the valuable comments and suggestions that has greatly helped in improving the manuscript. The manuscript was revised according to the reviewer’s comments and the authors are hopeful that the changes made especially in the methodology and results section have answered the concerns raised.
- Title:
1a) please include the type of study carried out
Author Response: The type of study has been included in the title of the revised manuscript.
Introduction:
1b) Page 2. Lines 44 - 48. References should be included to support the statements
Author Response: The authors thank the reviewer for this suggestion and suitable references has been included to support the mentioned statements.
1c) The authors justify the lack of evidence for the study in adults:
“However, to the best of our knowledge, there is no study on adults in Saudi Arabia that reports the basic knowledge, awareness and attitude towards the role of dietary fibers in a balanced diet and its impact on the health status”.
Nevertheless, the authors have included young people (20 years) in the sample. Could you explain why?
Author Response: The authors agree with the reviewer that young adults have also been included in this study. This was primarily as a result of the inclusion criteria mentioned in the “Methods” section (“healthy Saudi adult males and females aged between 20-70 years”). However, most of the study respondents were from 25-65 age-group (Table 1, 87.1% overall) and hence this study is mostly a representation of this adult Saudi age group.
- Materials and Methods
The information included in this section is very brief and does not allow the study to be reproduced. Authors should provide more information on the following issues:
2a) The authors should report more information on the sample. Type of sampling, sample size and inclusion/exclusion criteria. In addition, the results section should show the number of participants recruited and how many of them were excluded from the study.
Author Response: The authors thank the reviewer for this suggestion. The entire methodology section is revisited and revised to accommodate the concerns raised by the reviewer. Type of sampling, sample size, inclusion/exclusion criteria has been defined in the revised draft. Also, in the revised methodology section, it is mentioned that “response data from all these respondents were compiled, analysed and presented in this study and the subjects who were excluded were not interviewed”
2b) The authors should show more information about the validation and reliability process (items). Also, authors should include the questionnaire as supplementary material. In addition, in the results section, should show the reliability and validity data of the questionnaire and the modifications proposed by the experts.
Author Response: The authors thank the reviewer for this suggestion. The validated and approved version of the questionnaire that was used for the survey is included as the supplementary file. The revised section# 2.2 gives the details of validation and reliability of the sections of the questionnaire used in this survey. However, the authors feel that it will be cumbersome to detail all the modifications done to develop this questionnaire and it may divert the readers from the main objective of the study.
2c) I think that the statistical analysis is adequate and allows to answer the research question.
Author Response: The authors thank the reviewer for this appreciation.
- Results
3a) The results section is too long. I encourage authors not to repeat the information reported in the table.
Author Response: The authors thank the reviewer for this suggestion. The textual part in the results section is minimized extensively in the revised manuscript.
3b) I think it would be interesting to include in the table the statistically significant differences between genders of the characteristics shown in Table 1.
Author Response: The authors agree with the reviewer. The differences between genders were calculated by Pearson Chi-square test and depicted as p-values in table 1.
3c) In line with previous comments (methods). I don't understand the concept rarely (Table 2). When is it considered rarely, once a week, once a month... Please, this answer to the questionnaire must be explained in the methodology.
Author Response: The authors thank the reviewer for raising this question. The description of the questions asked in this section of the questionnaire was explained in more detail in “Data Collection and Measurement” section (2nd of the five parts of Questionnaire). “Thrice or more times per week” was classified as regular consumption; while anything less than that was classified as “rarely”. This has also been mentioned in the footnotes of the table for more clarity.
- Discussion
4a) Line 4. Please, change correlation by association
Author Response: This has been changed in the revised draft.
4b) Line 41. Please change; to.
Author Response: This has been changed in the revised draft.
4c) Line 62. Authors should unify sexes or gender throughout the text.
Author Response: ‘Sexes’ has been replaced with ‘gender’ throughout the manuscript. Thanks.
4d) Although the authors have acknowledged the socio-demographic differences as a limitation of the study, I think that a logistic regression should have been conducted to control for these differences. Please let me know your opinion.
Author Response: The authors thank the reviewer for this suggestion. The authors would like to assure that the multinomial regression results presented in the table 5 were already adjusted for socio-demographic differences between the genders and the adjusted p-value for the trend (pa) was depicted and this has been mentioned in the footnotes of the table. Also, the individual O.R.’s for each quartile against the lowest quartile (reference) were depicted as significant by * (p<0.05) or ** (p<0.01). This has been added in the footnotes as well.
Round 2
Reviewer 3 Report
Thank you for your diligent revisions. I believe that these have greatly improved the manuscript. However, I am attaching some comments between the lines.
Title:
1a) please include the type of study carried out
Author Response: The type of study has been included in the title of the revised manuscript.
R. The type of study is: cross-sectional study. Authors should change to design that they have indicated in the design
Introduction:
1b) Page 2. Lines 44 - 48. References should be included to support the statements
Author Response: The authors thank the reviewer for this suggestion and suitable references has been included to support the mentioned statements.
R. Thank you for considering the suggestion
1c) The authors justify the lack of evidence for the study in adults:
“However, to the best of our knowledge, there is no study on adults in Saudi Arabia that reports the basic knowledge, awareness and attitude towards the role of dietary fibers in a balanced diet and its impact on the health status”.
Nevertheless, the authors have included young people (20 years) in the sample. Could you explain why?
Author Response: The authors agree with the reviewer that young adults have also been included in this study. This was primarily as a result of the inclusion criteria mentioned in the “Methods” section (“healthy Saudi adult males and females aged between 20-70 years”). However, most of the study respondents were from 25-65 age-group (Table 1, 87.1% overall) and hence this study is mostly a representation of this adult Saudi age group.
R. I understand this is an inclusionary criterion. However, authors must be consistent. If the authors justify the research problem by not finding studies in adults, the inclusion criteria should have included only adults. I suggest the authors modify the introduction
- Materials and Methods
The information included in this section is very brief and does not allow the study to be reproduced. Authors should provide more information on the following issues:
2a) The authors should report more information on the sample. Type of sampling, sample size and inclusion/exclusion criteria. In addition, the results section should show the number of participants recruited and how many of them were excluded from the study.
Author Response: The authors thank the reviewer for this suggestion. The entire methodology section is revisited and revised to accommodate the concerns raised by the reviewer. Type of sampling, sample size, inclusion/exclusion criteria has been defined in the revised draft. Also, in the revised methodology section, it is mentioned that “response data from all these respondents were compiled, analysed and presented in this study and the subjects who were excluded were not interviewed”
R. Thank you for considering the suggestion. However, I have some questions:
I think there is a problem with the following answers:
"less than once a week", "1-2 times a week"
If it's less than once a week, they don't consume. I think the item is unclear.
2b) The authors should show more information about the validation and reliability process (items). Also, authors should include the questionnaire as supplementary material. In addition, in the results section, should show the reliability and validity data of the questionnaire and the modifications proposed by the experts.
Author Response: The authors thank the reviewer for this suggestion. The validated and approved version of the questionnaire that was used for the survey is included as the supplementary file. The revised section# 2.2 gives the details of validation and reliability of the sections of the questionnaire used in this survey. However, the authors feel that it will be cumbersome to detail all the modifications done to develop this questionnaire and it may divert the readers from the main objective of the study.
R. I don't agree with the authors. I think the study would be more valid if the authors included the statistical data on the validity and reliability of the questionnaire
2c) I think that the statistical analysis is adequate and allows to answer the research question.
Author Response: The authors thank the reviewer for this appreciation.
- Results
3a) The results section is too long. I encourage authors not to repeat the information reported in the table.
Author Response: The authors thank the reviewer for this suggestion. The textual part in the results section is minimized extensively in the revised manuscript.
R. Thank you for considering the suggestion
3b) I think it would be interesting to include in the table the statistically significant differences between genders of the characteristics shown in Table 1.
Author Response: The authors agree with the reviewer. The differences between genders were calculated by Pearson Chi-square test and depicted as p-values in table 1.
R. Please include in the statistical analysis section the tests used for table 1.
3c) In line with previous comments (methods). I don't understand the concept rarely (Table 2). When is it considered rarely, once a week, once a month... Please, this answer to the questionnaire must be explained in the methodology.
Author Response: The authors thank the reviewer for raising this question. The description of the questions asked in this section of the questionnaire was explained in more detail in “Data Collection and Measurement” section (2nd of the five parts of Questionnaire). “Thrice or more times per week” was classified as regular consumption; while anything less than that was classified as “rarely”. This has also been mentioned in the footnotes of the table for more clarity.
R. The authors have answered to all comments.
- Discussion
4a) Line 4. Please, change correlation by association
Author Response: This has been changed in the revised draft.
4b) Line 41. Please change; to.
Author Response: This has been changed in the revised draft.
4c) Line 62. Authors should unify sexes or gender throughout the text.
Author Response: ‘Sexes’ has been replaced with ‘gender’ throughout the manuscript. Thanks.
4d) Although the authors have acknowledged the socio-demographic differences as a limitation of the study, I think that a logistic regression should have been conducted to control for these differences. Please let me know your opinion.
Author Response: The authors thank the reviewer for this suggestion. The authors would like to assure that the multinomial regression results presented in the table 5 were already adjusted for socio-demographic differences between the genders and the adjusted p-value for the trend (pa) was depicted and this has been mentioned in the footnotes of the table. Also, the individual O.R.’s for each quartile against the lowest quartile (reference) were depicted as significant by * (p<0.05) or ** (p<0.01). This has been added in the footnotes as well.
R. Thank you for considering the suggestion
Author Response
Please see the attachment.

This manuscript is a resubmission of an earlier submission. The following is a list of the peer review reports and author responses from that submission.
Round 1
Reviewer 1 Report
The authors investigate the awareness, knowledge and consumption of dietary fiber in adult Arab population. The study is interesting and well-conducted, however it would have been worthwhile to distinguish between dietary fibers sources and types. Indeed, not all dietary fibers have the physiological effects presented in the introduction (lines 56 to 62) as these depend on their physico-chemical properties (solubility, fermentability and viscosity for instance) governed by their type and structure (e.g. pectins, hemicelluloses such as beta-glucans).
Lines 70-71: “higher price along with un-availability of varied range of cereal products”, this is not true everywhere in the world. In most African countries, high sources of dietary fibres (pseudo-cereals, fruit and vegetables, and legumes) are cheaper and widely available, particularly in rural areas. In Europe, India and Bangladesh, legumes (e.g. lentils, chickpeas, beans) are cheap and widely available as well. This statement needs to be rephrased.
Line 94: Given that 81.2% of respondents are female how representative is this study to the general population of Saudi Arabia?
Section 2.2. Who are those experts/reviewers? A bit more details (background, area of expertise, location) would be useful.
Lines 148-149: Oat is neither a dry fruit nor a legume.
Table 2: “Burghul” is wheat! The list lacks some sources of dietary fibre. What about other cereals such as rice? Breakfast cereals are not consumed in Saudi Arabia? What about other wheat- or cereal-derived products (e.g. pasta)?
Oat is not a legume but a cereal. The authors seem to have issue classifying plant-based foods.
Given the wide variety of fibers contained in vegetables (physicochemical properties and thereby effects on health), it would have made sense to divide vegetables into sub-categories (e.g. leafy, tubers and roots, cucurbits or squash, crucifers or Brassicas, etc…). The same apply to fruits (e.g. citrus rich in pectins).
Other minor remarks:
Line 20: “Apparently”, it does not seem to be needed here.
Line 23: add “the” before “…role of fiber rich foods”
Line 30: “have” instead of “having”, and “were not” instead of “not being”
Line 56: “…a highER dietary fiber intake than recommended…”
Line 57: “…abnormal distension…” of the abdomen?
Avoid starting a sentence by a figure/number such as lines 162 and 168 (59.4% and 47.8%), write it out in words.
Table 4: Last row, replace “…DF’s are easily not available” by “… DFs are not easily available”
Reviewer 2 Report
Dear authors,
The data reported in this manuscript show the results of the study on knowledge and habits of dietary fiber intake and its relationship with self-reported health status among Saudi adult population. These data on consumers perceptions are also considered important for the development of scientific knowledge.
However, I think that methodologically the study does not show enough quality to offer data robust enough to draw the conclusions.
According to the described methodology, the sample seems to have a significant bias, as its authors know. The authors do not specify in detail how the recruitment was, if any type of representability was followed or individual quotas were defined. If this were not so, the authors should justify it. If intended, these should justify too.
Regarding the data collected, it is not clear what type of previously validated or recognized questionnaires it has been based on, or how it has been the validation process mentioned that carried out. No bibliographic reference is mentioned in this regard. Perhaps the questionnaire should be attached to the manuscript.
In my view, the results could not be interpreted appropriately due to being carried out with a non-representative sample and a questionnaire that could be conditioned. Perhaps it is too hasty to draw conclusions through odds ratio analysis.
Kind regards,